# Damage and Technical Wear of Tenement Houses in Fuzzy Set Categories

Jarosław Konior *, Marek Sawicki and Mariusz Szóstak

Department of Building Engineering, Faculty of Civil Engineering, Wroclaw University of Science and Technology, 50-370 Wrocław, Poland; marek.sawicki@pwr.edu.pl (M.S.); mariusz.szostak@pwr.edu.pl (M.S.)
* Correspondence: jaroslaw.konior@pwr.edu.pl; Tel.: +48-71-320-23-69

**Abstract:** The results and conclusions of the research presented in the article concern the topic of the technical maintenance and wear of traditionally erected residential buildings. The cause and effect relations between the occurrence of damage to the elements of tenement houses, which are treated as an expression of their maintenance conditions, and the size of the technical wear of these elements were determined in a representative and purposefully selected sample of 102 apartment houses built in the second half of the 19th and early 20th centuries in the Wroclaw, Poland downtown district "Srodmiescie". Recognition of the impact of the maintenance of residential buildings on the level of their technical wear was carried out using quantitative methods from fuzzy set categories, and also with the use of the authors' own model. The created model, based on the Zadeh function, was created in fuzzy conditions for the purpose of assessing the degree of damage to selected building elements. The treatment of the problem with regard to fuzzy criteria allowed for the synthesis of elementary criteria, which give the greatest approximations at the technical research stage of a residential building, into a global assessment of the degree of the wear of its elements. Moreover, it also significantly reduced the subjective factor of this assessment, which had a significant impact on the results of the research obtained in the case of good, medium and poor conditions of tenement houses. It was proven that the conditions of maintenance and use of buildings determine the amount of technical wear of their elements. The state of exploitation of the examined tenement houses is reflected in the mechanical damage to the internal structure of the elements (determined in fuzzy categories). This damage has a significant frequency and cumulative effects, and is characteristic for buildings with satisfactory and average maintenance.

**Keywords:** tenement houses; technical wear; damage; maintenance; fuzzy sets

## 1. Introduction

### 1.1. Source Literature

The aim of the research was to identify the impact of the processes associated with the broadly understood maintenance of old residential buildings with a traditional construction on the size and intensity of the wear of their elements. The degree of technical wear of residential building elements is a parameter of fundamental importance in the comprehensive assessment of their technical condition, regardless of the approach that was used in the test method. The aim of the research was achieved through the analysis of the symptoms of the technical wear process—understanding the mechanism of the phenomenon of damage and identifying the size and intensity of damage to the elements of the evaluated buildings.

Essential research of tenement houses aims to undertake a qualitative analysis of detected defects and identify all particular defects of their elements. Therefore, the "reason–effect" model is applied as follows:

$$[\text{REASONS}] \rightarrow [\text{observed} \xleftrightarrow{\text{SYMPTOMS}} \text{measured}] \rightarrow [\text{EFFECTS}]$$

Commonly used mathematical methods and the broadly understood system analysis deal with real tasks in which the basic goal is the possibility of including all the types of indeterminacy among modeled quantities and the relationships between them. Every indeterminacy has traditionally been equated with the uncertainty of a random type, which has enabled known probabilistic and statistical tools to be used. In practice, however, there are many cases in which the indeterminacy of the type of inaccuracy, ambiguity and imprecision of meanings can be found. However, these situations are not of a random nature, and therefore traditional probabilistic models may not be adequate [1–7]. When assessing the possibility of random and/or fuzzy events occurring in construction investment projects, apart from immeasurable (qualitative) criteria, measurable (quantitative) criteria are also used. These quantitative criteria are expressed in a mathematical model that describes multiple phenomena of construction engineering processes. Only some of these criteria are strictly defined concepts—boundary, extreme. Most of these criteria are approximate. Their value is determined using descriptive methods, e.g., "good quality", "short term", "low budget". Therefore, concepts of this type cannot be adequately represented as a conventional set. To overcome this difficulty, in 1965, *Lotfi A. Zadeh* of the University of California in Berkeley introduced the concept of a fuzzy set with its membership function [8–10].

*Zadeh* [8,9], when developing the foundations of fuzzy set theory, formulated the following principle: "in general, complexity and precision are inversely related to each other in the sense that if the complexity of the problem under consideration increases, the possibility of its precise analysis decreases". *Yager* [11,12] independently came to a similar conclusion when examining the uncertainty in probability. However, people can cope with situations in which all attempts at the mathematical formalization of a task and its solutions are unsuccessful due to the fact that, e.g., it is impossible to build an exact mathematical model, or it would take too long to solve it. *Zadeh* saw the reasons for this in the ability of the human mind to think in approximate categories, which microprocessors do not have. Due to this, a person can process approximate and ambiguous data, create models of the most complex processes, determine approximate solutions, etc. Fuzzy set theory, according to *Zadeh* and *Yager*, is therefore a tool used to formalize this approximate reasoning in vague and ambiguous terms.

For a long time, "uncertainty" and "ambiguity" have been used as synonyms for a lack of knowledge, which is decreasing as research progresses. Relatively recently, starting from the 1970s, these terms began to be treated as a reflection of reality, without the previous clearly negative meaning. It was then that the first major works in the field of multiple applications of fuzzy sets occurred, including *Zadeh* [8,9], *Yager* [11,12] and *Sanchez* [13]. Summing up, among the formal apparatuses that led to the development of fuzzy set theory, the first place is occupied by multi-valued logics. Previously, since ancient times, almost the entire development of logic could have been equated with two-valued logic, in which a statement can only be either true or false. The fact of such a polarization of truth and falsehood was considered as an essential feature of any "logical" reasoning. Many logicians, represented especially by *Lukaszewicz* (a co-founder of the Polish School of Logic), were already aware of the mismatch between such "rigid" logic and reality. The explosion of interest in multi-valued logics also aroused a significant increase in the interest of fuzziness and its origins, which was widely described in the later works of *Zadeh* [9], *Yager* [12], *Sanchez* [13] and *Kasprzyk* [14].

The stage preceding the main scope of the work was the conducting of a qualitative analysis of damage to the elements of the tested residential buildings [1,15]. The technical characteristics and typological ordering of this damage, understood as an expression of the quality of maintenance of residential buildings, enabled the exploitation conditions of the considered objects to be identified.

A number of works by *Nowogońska* [16–20] were used in the methodical approach to the technical assessment of tenement houses, and the fuzzy calculus presented in the publications of *Plebankiewicz, Wieczorek* and *Zima* [21–26] was used in the assessment of the whole service life of a building object. The works of *Ibadov* [27–30] and other authors [31–39],

which concerned the construction investment process with the fuzzy phase, allowed for the practical application of uncertain and subjective events when determining the degree of damage to the tested tenement houses.

### 1.2. Subject of Study

A group of old tenement houses (that is, those erected before the First World War) takes an important place in Polish building resources. This group includes about 10.1% of the whole number of urban flats. What is more, the importance of this type of building relies on the fact that it takes part in creating an urban environment. At present, an action needs to be directed to the repair of the old land development. Doubtless, cultural aspects motivate all this action. To estimate its technical and economic justification, the degree of the technical wear of the old land development must be recognized and calculated.

This paper is a result of technical research and analyses on the old apartment houses in Wrocław, Poland [40]. The aim of the analysis is to provide information, which should help to direct an action, connected with this group of residential buildings. They are the apartment houses which were built at the turn of the nineteenth and twentieth centuries. The buildings are situated in the part of the city which (as a district from very few ones) was not completely destroyed by the war activities. The apartment houses are three- or four-storey buildings, made of bricks, erected in longitudinal, usually three-row, structural systems. Apart from the floors over the basement, which are solid ones, all the inter-storey floors represent typical wooden floors. All the buildings are covered with wooden rafter framing, usually a purlin–collar one. The staircases are composed of wooden or steel structural elements with wooden flights of steps.

### 1.3. Research Problem

While appraising building elements' technical wear—apart from applying the measurable (qualitative) criteria—the immeasurable (quantitative) criteria representing symptoms (pinpointed defects) of their deterioration have been taken into account. Only very few of these criteria can be classified at a high level of probability. There are symptoms of extreme characters, described by extreme dichotomic divisions. It is, however, agreed that between, e.g., a total pest attack to wooden elements and a lack of pests, the mid-states appear. Their value is often appreciated in a verbal way, e.g., "substantially", considerably", "significantly", "partially", "hardly" and it is always used in a description of detected defects as a result of a building object's technical inspections.

When assessing the degree of technical wear of building elements, apart from measurable (quantitative) criteria, immeasurable (qualitative) criteria are also used. They are expressed in the analysis of symptoms, i.e., damage, which lowers the technical condition and utility value of building elements. Only some of these criteria can be quantified with a big approximation. These are the symptoms with an extreme character, e.g., inter-story ceilings that are replaced with new elements that are not damp. It can then be assumed that the damage, and the technical wear it causes, take a value of zero.

In turn, flooding of the floors above basements does not raise doubts regarding the occurrence of the total dampness, and therefore the degrees of damage and technical wear caused by moisture take values equal to one within the variability interval of [0, 1]. Most of these criteria, however, are qualitative. Their value is determined verbally, e.g., as "significant", "poor", "strong", "almost not at all", "partial" or "complete", and it always appears in the description of damage phenomena. The interpretation of the effects of these phenomena, which is performed according to qualitative (i.e., subjective) premises, leads to the indiscriminate categorization of the technical maintenance conditions for buildings and their elements, i.e., good, satisfactory, average, poor or bad. Therefore, can a building element with a degree of technical wear of, e.g., 15%, be considered good or satisfactory from the point of view of the technical maintenance quality? Does significant biological contamination of wooden floor beams determine their 100% wear?

Striving for a quantification of criteria that are inherently qualitative (and therefore immeasurable), and trying to determine the relations between them, led to the use of the category of fuzzy sets with regard to this issue. Their properties enable damage to building elements, as well as the conditions of their technical maintenance, to be described within an unambiguous quantitative (measurable) aspect.

Therefore, the research led towards looking at the problem from this angle, which allowed the description of naturally qualitative (immeasurable) variables and the determination of existing relations between them in fuzzy set categories [15,25–39]. The advantages of fuzzy theory made it possible to describe the defects, representing three middle states (II, III, IV) of conditions of the building elements' maintenance, in a clear quantitative (measurable) aspect. Doubtless, fuzzy conditions are fully represented in these mid-states.

## 2. Research Methodology

### 2.1. Fuzzy Set Theory

The basic concept of the theory that was used in this paper is the concept of a fuzzy set [8,9,11–14]. The definition of a fuzzy set can be formulated as follows: a fuzzy set is set A, the x elements of which are characterized by the lack of a clear boundary between the membership and non-membership of x to A. The degree of the membership of element x to fuzzy set A is described by function $\mu A(x)$, which is called the membership function. The $\mu A(x)$ function takes values from the interval of [0, 1], where:

$\mu A(x) = 0$, which means that x is not a member of A;

$\mu A(x) = 1$, which means that x is a full member of A.

Fuzzy set A in a certain space (in this paper, it is the area of considerations concerning the observed states) $X = \{x\}$, which is written as $A \subseteq X$, is called the set of pairs:

$$A = \{(\mu A(x), x)\}, \forall\, x \in X.$$

Therefore, two basic fuzzy sets can be distinguished in a problem (each one is described in the three following observed states—II, III, IV):

- a fuzzy set of the technical wear of building elements $A \subseteq Ze \Leftrightarrow Z$ (to simplify the designations): $Z = \{(\mu Z(z), z)\}, \forall\, z \in Z$;
- a fuzzy set of damage to building elements $B \subseteq U$:
  $U = \{(\mu U(u), u)\}, \forall\, u \in U$.

The basic operations performed on the fuzzy sets defined in the article are presented below:

- the absolute complement of the fuzzy set $A \subseteq X$, denoted as $-A$:

$$\mu_{-A}(x) = 1 - \mu_A(x), \forall\, x \in X \tag{1}$$

- the multiple sum of fuzzy sets $A, B \subseteq X$, denoted as $A \cup B$:

$$\mu_{A \cup B}(x) = \mu_A(x) \vee \mu_B(x), \forall\, x \in X \text{ (symbol } \vee \text{ denotes „max")} \tag{2}$$

- the intersection of fuzzy sets $A, B \subseteq X$, denoted as $A \cap B$:

$$\mu_{A \cap B}(x) = \mu_A(x) \wedge \mu_B(x), \forall\, x \in X \text{ (symbol } \wedge \text{ denotes „min")} \tag{3}$$

- the k-th power (k > 0) of fuzzy set $A \subseteq X$, denoted as Ak:

$$\mu_A{}^k(x) = (\mu(x))^k, \forall\, x \in X \tag{4}$$

Special cases of exponentiation include:

- the concentration of fuzzy set A ⊆ X, denoted as CON (A):

$$\mu_{CON(A)}(x) = (\mu_A(x))^2, \forall\, x \in X \tag{5}$$

- the dilution of fuzzy set A ⊆ X, denoted as DIL (A):

$$\mu_{DIL(A)}(x) = (\mu_A(x))^{0.5}, \forall\, x \in X \tag{6}$$

All these operations, which are of great importance in linguistic semantics, are interpreted as:

- −A ⇔ "not A";
- A ∪ B ⇔ "A or B";
- A ∩ B ⇔ "A and B";
- CON (A) ⇔ "strong A" (crispens the fuzzy set);
- DIL (A) ⇔ "more or less, likely A" (flattens the fuzzy set).

When visually assessing the technical wear of building elements that inspected tenement houses consist of, the symptoms of their damage are taken into account, i.e., individual damage that can be categorized into the following groups of damage:

- UM—mechanical damage to the structure and texture of building elements;
- UW—damage to building elements caused by water penetration and moisture penetration;
- UD—damage resulting from the loss of the original shape of wooden elements;
- UP—damage to wooden elements attacked by biological pests.

The purpose of such a conceptual and technical systematization of damage is a comprehensive diagnosis of the extent to which a building element is worn. This assessment, in turn, leads to the implication of stating under what technical conditions—good, satisfactory, average, poor or bad—the building element was (is) maintained. The terms "good technical condition of maintenance", "satisfactory technical condition of maintenance", etc., can be considered as fuzzy sets with regard to semantic (qualitative) and technical (quantitative) aspects.

It is difficult to define a fuzzy set with such a broad meaning as "average technical condition of maintenance" using one membership function. In this case, a semantic analysis of the term "technical wear of a building element" was used, which was denoted with the symbol of a fuzzy set "Z". Let the technical wear of building element Z consist of: mechanical wear of its structure and texture (fuzzy set ZM), its technical wear caused by water penetration and moisture penetration (fuzzy set ZW), technical wear resulting from the loss of its original shape (fuzzy set ZD) and technical wear caused by the attack of biological pests (fuzzy ZP harvest). This sum can then be expressed as follows:

$$Z = ZM \cup ZW \cup ZD \cup ZP \tag{7}$$

and when assuming the identity of the degree of technical wear and its visual symptom (Z ⇔ U)—damage to a building element that is integrated into the above-described damage sets (Expression (7))—it takes the following form:

$$U = UM \cup UW \cup UD \cup UP \tag{8}$$

### 2.2. Research Model

The aim of the proposed model is to assess the technical wear of a building element with regard to the overriding criterion, i.e., "slightly worn, worn, significantly worn". The concepts defined in this way at the basic level best describe the behavior of a building element in its three middle maintenance states. It is in them, after rejecting the extreme states (i.e., good and bad) that have the most reliable evaluation principles from the technical point of view, that the fuzzy conditions are most fully represented. The basic

principles of fuzzy logic and approximate reasoning were applied [8,9,11–15], and the fuzzy state was described as follows: its damage means that it can be classified as being in a satisfactory (II), average (III) and poor (IV) technical condition of maintenance. Therefore, in each maintenance state, the total damage to a building element is a multiplicity sum of the sets of damage, and it is expressed by Formula (9):

$$U(II, III, IV) = UM(II, III, IV) \cup UW(II, III, IV) \cup UD(II, III, IV) \cup UP(II, III, IV) \quad (9)$$

where each set of damage, in each of the three maintenance states (II, III, IV), is a set of basic damage $u_j$, which represents elementary lower order criteria:

- UM = $\{u_1, u_2, \dots, u_{14}\}$;
- UW = $\{u_{15}, u_{16}, \dots, u_{23}\}$;
- UD = $\{u_{24}, u_{25}, \dots, u_{28}\}$;
- UP = $\{u_{29}, u_{30}\}$.

Multiplication sum (9) can be written in each of the three maintenance states (II, III, IV) using the membership function:

$$\mu_U = \mu_{UM} \vee \mu_{UW} \vee \mu_{UD} \vee \mu_{UP} \quad (10)$$

There is an intermediate stage between identifying damage at the elementary level, which occurs in everyday construction practice, and merging it into sets of damage in terms of their similarity regarding the wear processes. This stage involves the selection of damage of the same type but of different intensity (e.g., pitting corrosion, surface corrosion, deep corrosion of steel beams), or damage occurring to complex elements (e.g., structural walls—decay of brick or mortar). This method of combining elementary damage was used in the research, which led to the obtaining of greater possibilities of using operations of system analysis in fuzzy sets. In the considered sample of downtown tenement houses, this division is as follows:

- $\{u_1, u_2\}$ = U1 ⇔ mechanical damage and leaks;
- $\{u_3, u_4\}$ = U2 ⇔ brick and mortar losses;
- $\{u_5, u_6\}$ = U3 ⇔ brick and mortar decay;
- $\{u_7, u_8\}$ = U4 ⇔ peeling off and decomposing of the paint coatings;
- $\{u_9, u_{10}\}$ = U5 ⇔ cracks in brick and plaster;
- $\{u_{11}, u_{12}\}$ = U6 ⇔ scratching on walls and plaster;
- $\{u_{13}, u_{14}\}$ = U7 ⇔ loosening and falling off of plaster sheets;
- $\{u_{15}, u_{16}, u_{23}\}$ = U8 ⇔ dampness, weeping and flooding with water;
- $\{u_{17}, u_{18}, u_{19}\}$ = U9 ⇔ brick corrosion, fungus and mold;
- $\{u_{20}, u_{21}, u_{22}\}$ = U10 ⇔ pitting corrosion, surface corrosion and deep corrosion of steel beams;
- $\{u_{24}, u_{25}\}$ = U12 ⇔ dynamic sensitivity and deformation of floor beams;
- $\{u_{26}, u_{27}, u_{28}\}$ = U13 ⇔ torsional buckling and distortion of window joinery and wood elements;
- $\{u_{29}, u_{30}\}$ = U14 ⇔ touchwood and biological infestation of wooden elements.

In each of the damage types distinguished in this way, there is an intersection of two or three elementary fuzzy sets. Between them, as is the case between sets of damage, there is a multiple sum of the fuzzy sets that are defined above. All these dependencies can be described by the general formula for assessing the degree of damage to the elements of the analyzed residential buildings in their middle maintenance states which, when using the membership function, is as follows:

$$\mu_U = (\mu_{u1} \wedge \mu_{u2}) \vee (\mu_{u3} \wedge \mu_{u4}) \vee (\mu_{u5} \wedge \mu_{u6}) \vee (\mu_{u7} \wedge \mu_{u8}) \vee (\mu_{u9} \wedge \mu_{u10}) \vee$$
$$\vee (\mu_{u11} \wedge \mu_{u12}) \vee (\mu_{u13} \wedge \mu_{u14}) \vee (\mu_{u15} \wedge \mu_{u16} \wedge \mu_{u23}) \vee (\mu_{u17} \wedge \mu_{u18} \wedge \mu_{u19}) \vee \quad (11)$$
$$\vee (\mu_{u20} \wedge \mu_{u21} \wedge \mu_{u22}) \vee (\mu_{u24} \wedge \mu_{u25}) \vee (\mu_{u26} \wedge \mu_{u27} \wedge \mu_{u28}) \vee (\mu_{u29} \wedge \mu_{u30})$$

Due to the fact that the greatest approximations of the observed states can be obtained at the level of elementary criteria, the degrees of membership of damage $u_1 \div u_{30}$ to fuzzy sets UM, UW, UD, UP were calculated at the stage of the basic comparative analysis, in which the fundamental probabilistic measure is the probability of the occurrence of a single damage $p(u_j)$ in the II, III and IV maintenance states. The probability of $p(u_j)$ is therefore a feature that determines the membership to elementary sets $u_1 \div u_{30}$. It would not be a mistake to simply identify the probabilities $p(u_j)$ with the degrees of memberships $\mu_{uj}$, which are described linearly by the membership function operating on the domain [0, 1]. However, in order to present the properties of fuzzy sets more closely, the function used by *Zadeh* [8–10] was chosen for intensifying the contrast of the fuzzy set $A \subseteq X$:

$$\mu_{INT(A)}(x) = \begin{cases} 2(\mu_A(x))^2, \forall x : \mu_A(x) < 0.5 \\ 1 - 2(1 - \mu_A(x))^2, \forall x : \mu_A(x) \geq 0.5 \end{cases} \tag{12}$$

Therefore, the intensification of contrast increases the membership degrees that are greater than or equal to 0.5, while reducing the membership degrees that are lower than 0.5 (Figure 1).

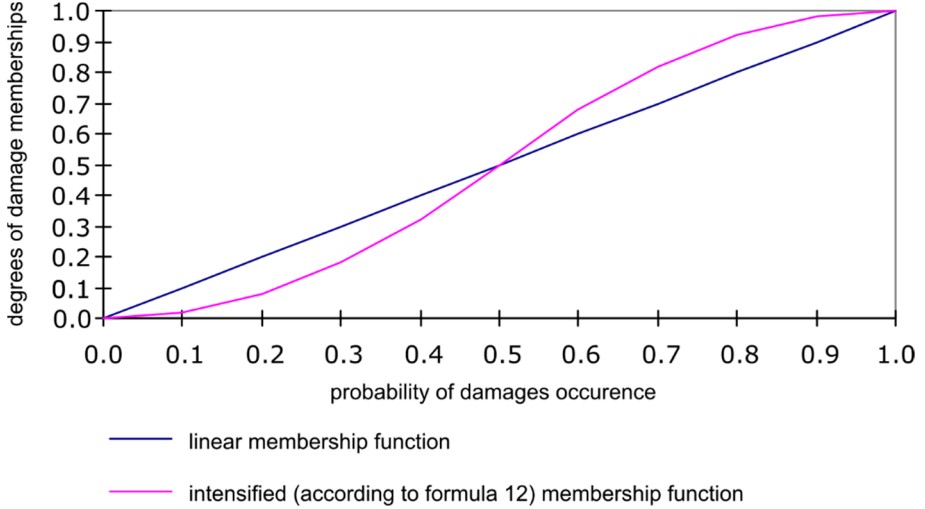

**Figure 1.** The effect of the contrast intensification of the degrees of damage memberships.

The final stage of the created model for assessing the technical wear (damage degree) of selected building elements in the three middle states of their technical maintenance is to estimate the size of the impact of elementary damage on the total damage. The study of the observed states and the conclusions from the proposed method of associating the occurring damage with the occurrence of the process of technical wear indicate a significant range of the strength of this relation within one building element in the maintenance states II, III and IV [1–7]. None of the values of the bi-serial correlation coefficient $r(Z)$, which is a measure of this relationship, reaches a value of 1 in domain [0, 1]. Therefore, when taking the extreme value from this range as a reference point, it can be assumed that none of the values of the correlation coefficient $r(Z)$ concentrates the fuzzy set U, while each of them—to a different degree—dilutes it. Considerations regarding the relationship between these dependencies and the analysis of the effects of the dilution process of fuzzy sets have led to the determination of the weights of the degrees of membership of elementary damage uj as a function of the correlation coefficient $r(Z)$:

$$\mu_{uj} = [f(\mu_{uj})]^{1/r(Z)} \tag{13}$$

The result of the proposed Formula (13) is the following change in the membership function:

- $r(Z) \rightarrow 0 \Rightarrow \mu_{uj} \rightarrow 0$;
- $r(Z) \rightarrow 1 \Rightarrow \mu_{uj} \rightarrow \mu_{uj}$.

The application of the original procedures of the intensification and dilution of membership functions, according to Formulas (12) and (13), to the general Formula (11) of the model for assessing the degree of damage to the elements of the analyzed tenement houses in terms of fuzzy sets allowed for the transition from the data recorded using non-measurable variables to results defined by measurable values. The proposed model gives a numerical answer to the question of to what extent is a building element damaged. The total degrees of damage to the ten selected elements of the analyzed buildings S(U) in the maintenance states II, III and IV are presented in Table 1.

**Table 1.** The degree of fuzzy damage to building elements in their middle maintenance states. (grey backgroud is necessary to distinguish extreme values)

| Group Number | Building Element | Damage Number | Damage Description | Degree of Fuzzy Damage Set S(U) Corresponding to the Maintenance States II, III and IV | | |
|---|---|---|---|---|---|---|
| | | | | S(U)II | S(U)III | S(U)IV |
| Z2 | Foundations | $u_3$ | brick losses | 0.24 | 0 | 0 |
| | | $u_5$ | brick decay | 0 | 0.59 | 0 |
| | | $u_9$ | brick cracks | 0 | 0 | 0 |
| | | $u_{15}$ | dampness of foundations | 0 | 0 | 0 |
| | | $u_{16}$ | weeping on foundations | 0 | 0 | 0.97 |
| | | $u_{17}$ | biological corrosion of bricks | 0 | 0 | 0 |
| | | $u_{19}$ | mold and rot on foundations | 0 | 0 | 0 |
| Z3 | Basement walls | $u_3$ | brick losses | 0.05 | 0.25 | 0.67 |
| | | $u_4$ | mortar losses | 0 | 0 | 0 |
| | | $u_5$ | brick decay | 0 | 0 | 0 |
| | | $u_6$ | mortar decay | 0 | 0 | 0 |
| | | $u_9$ | cracks in bricks | 0 | 0 | 0 |
| | | $u_{10}$ | cracks in mortar | 0 | 0 | 0 |
| | | $u_{15}$ | dampness of walls | 0 | 0 | 0 |
| | | $u_{16}$ | weeping on walls | 0 | 0 | 0 |
| | | $u_{17}$ | biological corrosion of bricks | 0 | 0 | 0 |
| | | $u_{19}$ | mold and rot on walls | 0 | 0 | 0 |
| Z4 | Solid floors above basements | $u_3$ | brick losses | 0.01 | 0.22 | 0 |
| | | $u_5$ | brick decay | 0 | 0 | 0 |
| | | $u_9$ | cracks in bricks | 0 | 0 | 0 |
| | | $u_{15}$ | dampness of floors | 0 | 0 | 0 |
| | | $u_{16}$ | weeping on floors | 0 | 0 | 0.50 |
| | | $u_{20}$ | corrosion raid on steel beams | 0 | 0 | 0 |
| | | $u_{21}$ | surface corrosion of steel beams | 0 | 0 | 0 |
| | | $u_{22}$ | deep corrosion of steel beams | 0 | 0 | 0 |
| | | $u_{23}$ | flooding of floors with water | 0 | 0 | 0 |

**Table 1.** *Cont.*

| Group Number | Building Element | Damage Number | Damage Description | Degree of Fuzzy Damage Set S(U) Corresponding to the Maintenance States II, III and IV | | |
|---|---|---|---|---|---|---|
| | | | | S(U)II | S(U)III | S(U)IV |
| Z7 | Structural walls | $u_3$ | brick losses | 0 | 0 | 1.00 |
| | | $u_4$ | mortar losses | 0.34 | 0.93 | 0 |
| | | $u_5$ | brick decay | 0 | 0 | 0 |
| | | $u_6$ | mortar decay | 0 | 0 | 0 |
| | | $u_9$ | cracks in bricks | 0 | 0 | 0 |
| | | $u_{10}$ | cracks on plaster | 0 | 0 | 0 |
| | | $u_{11}$ | scratching on walls | 0 | 0 | 0 |
| | | $u_{12}$ | scratching on plaster | 0 | 0 | 0 |
| | | $u_{15}$ | dampness of walls | 0 | 0 | 0 |
| | | $u_{16}$ | weeping on walls | 0 | 0 | 0 |
| | | $u_{17}$ | biological corrosion of bricks | 0 | 0 | 0 |
| | | $u_{19}$ | mold and rot on walls | 0 | 0 | 0 |
| Z8 | Inter-story wooden floors | $u_{12}$ | scratching on the plaster of the ceiling | 0 | 0 | 0 |
| | | $u_{13}$ | peeling of ceiling plaster | 0 | 0 | 0 |
| | | $u_{15}$ | dampness of floors | 0 | 0 | 0 |
| | | $u_{16}$ | weeping on floors | 0.01 | 0.64 | 0 |
| | | $u_{18}$ | fungus on floors | 0 | 0 | 0.49 |
| | | $u_{24}$ | dynamic sensitivity of floor beams | 0 | 0 | 0 |
| | | $u_{25}$ | deformations of wooden beams | 0 | 0 | 0 |
| | | $u_{30}$ | complete insect infestation of wooden beams | 0 | 0 | 0 |
| Z9 | Stairs | $u_1$ | mechanical damage | 0.26 | 0.56 | 0 |
| | | $u_3$ | brick losses | 0 | 0 | 0 |
| | | $u_{16}$ | weeping on stairs | 0 | 0 | 0.95 |
| | | $u_{20}$ | corrosion raid on steel beams | 0 | 0 | 0 |
| | | $u_{21}$ | surface corrosion of steel beams | 0 | 0 | 0 |
| | | $u_{22}$ | deep corrosion of steel beams | 0 | 0 | 0 |
| | | $u_{29}$ | partial insect infestation of wooden elements | 0 | 0 | 0 |
| Z10 | Roof construction | $u_{15}$ | dampness of truss | 0 | 0 | 0 |
| | | $u_{16}$ | weeping on wooden elements | 0 | 0.43 | 0.53 |
| | | $u_{28}$ | delamination of beams | 0.03 | 0 | 0 |
| | | $u_{29}$ | partial insect infestation of wooden elements | 0 | 0 | 0 |
| | | $u_{30}$ | complete insect infestation of wooden beams | 0 | 0 | 0 |

<div align="center">**Table 1.** *Cont.*</div>

| Group Number | Building Element | Damage Number | Damage Description | Degree of Fuzzy Damage Set S(U) Corresponding to the Maintenance States II, III and IV | | |
|---|---|---|---|---|---|---|
| | | | | S(U)II | S(U)III | S(U)IV |
| Z13 | Window joinery | $u_1$ | mechanical damage | 0 | 0.85 | 0 |
| | | $u_2$ | window leaks | 0.89 | 0 | 1.00 |
| | | $u_{15}$ | dampness of windows | 0 | 0 | 0 |
| | | $u_{16}$ | stains on windows | 0 | 0 | 0 |
| | | $u_{19}$ | mold and rot on windows | 0 | 0 | 0 |
| | | $u_{26}$ | skewing of window joinery | 0 | 0 | 0 |
| | | $u_{27}$ | warping of window joinery | 0 | 0 | 0 |
| | | $u_{29}$ | partial insect infestation of window joinery | 0 | 0 | 0 |
| | | $u_{30}$ | complete insect infestation of window joinery | 0 | 0 | 0 |
| Z15 | Inner plasters | $u_1$ | mechanical damage to plaster | 0.40 | 0 | 0 |
| | | $u_6$ | plaster decay | 0 | 0.85 | 0 |
| | | $u_7$ | peeling off of paint coatings | 0 | 0 | 0 |
| | | $u_8$ | falling off of paint coatings | 0 | 0 | 0 |
| | | $u_{10}$ | cracks in plaster | 0 | 0 | 0.95 |
| | | $u_{12}$ | scratching on plaster | 0 | 0 | 0 |
| | | $u_{13}$ | loosening of plaster | 0 | 0 | 0 |
| | | $u_{14}$ | flaking off of sheets of plaster | 0 | 0 | 0 |
| | | $u_{15}$ | dampness of plaster | 0 | 0 | 0 |
| | | $u_{16}$ | weeping on plaster | 0 | 0 | 0 |
| | | $u_{18}$ | fungus on plaster | 0 | 0 | 0 |
| | | $u_{19}$ | mold and rot on plaster | 0 | 0 | 0 |
| Z20 | Facades | $u_1$ | mechanical damage to plaster | 0 | 0 | 0 |
| | | $u_6$ | plaster decay | 0.43 | 0 | 0 |
| | | $u_7$ | peeling off of paint coatings | 0 | 0 | 0 |
| | | $u_8$ | falling off of paint coatings | 0 | 0 | 0 |
| | | $u_{10}$ | cracks in plaster | 0 | 0.94 | 0 |
| | | $u_{12}$ | scratching on plaster | 0 | 0 | 1.00 |
| | | $u_{13}$ | loosening of plaster | 0 | 0 | 0 |
| | | $u_{14}$ | flaking off of sheets of plaster | 0 | 0 | 0 |
| | | $u_{15}$ | dampness of plaster | 0 | 0 | 0 |
| | | $u_{16}$ | weeping on plaster | 0 | 0 | 0 |
| | | $u_{18}$ | fungus on plaster | 0 | 0 | 0 |
| | | $u_{19}$ | mold and rot on plaster | 0 | 0 | 0 |

## 3. Results

The analysis of the results of the research concerning the impact of damage to building elements on their technical wear with regard to fuzzy sets leads to the following conclusions (Table 1):

a.  in the field of assessing the degree of fuzzy damage to elements of downtown tenement houses—S(U) II, III, IV:

- the development of the model presented in the article allowed the fundamental question of to what extent a building element is worn (damaged), when knowing that it is (more or less) satisfactorily, moderately or poorly maintained, to be answered;
- the use of simple operations in the fuzzy set calculus enabled the influence of both elementary damage that occurs with a specific frequency (probability) and the measure of its interdependence (correlation) on the observed technical wear of building elements to be considered;
- as a result of the proposed model, which is based on fuzzy set theory, it was possible to identify the elementary damage that determines the degree of destruction of the building's elements;

b.  when determining the degree of damage of 10 selected building elements according to fuzzy criteria, it was indicated that there is a need for an individual approach to each of the elements (especially structural) during the process of their technical assessment. However, several regularities can be identified:

- the degree of damage to the element increases with the deterioration of its maintenance conditions (although not proportionally to the maintenance conditions and not equally for different types of elements). For instance, degrees of fuzzy damage set S(U) corresponding to the maintenance states II, III and IV grow in the following way: Z3—basement walls—$u_3$—brick losses: 0.05; 0.25; 0.67. It most often differs from the observed values of the degree of the technical wear that was determined using the probabilistic approach [1]—in particular, in poor conditions of building maintenance, the degree of damage exceeds 70% of its technical wear threshold;
- elementary damage that determines the degree of destruction of an element comes much more often from group I (mechanical damage to the structure and texture of elements) than was the case in the analysis of the observed states. Only under poor conditions of building maintenance does the analysis of the observed random [1] and fuzzy [15] phenomena show a great similarity—the decisive damage is the destruction of the element caused by water penetration and moisture penetration (group II);
- at the level of the greatest detail, the type of damage and the degrees of fuzzy damage to the elements of the downtown tenement houses were determined. In the most representative, i.e., average/satisfactory condition of maintenance—S (U) III—the degrees were as follows:
  ○  for foundations: brick decay 0.59
  ○  for basement walls: brick decrements 0.25
  ○  for solid floors above basements: brick decrements 0.22
  ○  for structural walls: mortar decrements 0.93
  ○  for wooden inter-storey floors: weeping 0.64
  ○  for internal stairs: mechanical damage 0.56
  ○  for roof constructions: weeping on wooden elements 0.43
  ○  for window joinery: mechanical damage 0.85
  ○  for inner plasters: plaster decay 0.85
  ○  for facades: cracks on plaster 0.94

## 4. Summary and Discussion

At the beginning, general methodological conclusions were formulated. They resulted from the modeling of the impact of the maintenance of tenement houses on the technical wear of their elements in fuzzy conditions. Such an approach gives much greater possibilities of studying cause and effect relationships than the probabilistic analysis [1]:

a. the use of simple operations in the fuzzy set calculus enables the simultaneous recognition of the impact of elementary damage that occurs with a specific frequency (probability), and also the measure of its interdependence (correlation) on the observed technical wear of building elements;

b. in the effect of fuzzy transformations, it is possible to identify the elementary damage that determines the degree of destruction of the building element. The result of the cumulative effects of frequently occurring mechanical damage to the structure and texture of elements indicates that this type of damage is no less important in the process of the technical wear of elements of downtown tenement houses;

c. consideration of the problem with regard to fuzzy phenomena allows for the synthesis of elementary criteria. This gives the greatest approximations (at the stage of the technical investigation of a residential building) for the global assessment of the degree of wear of the building's elements. In addition, it significantly reduces the subjective factor of this assessment, which has the greatest impact on the result of research conducted for the middle maintenance states of buildings.

The consequence of systematizing the most important processes that influence the loss of functional properties of residential buildings was the creation of the authors; own qualitative model and its transformation into a quantitative model. This, in turn, enabled a multi-criteria quantitative analysis of the cause–effect phenomena—"damage–technical wear"—of the most important elements of downtown residential buildings to be conducted in the so-called conventional and fuzzy sets. In conventional sets, in which attempts were made to describe the observed (empirical) states with the use of theoretical formulas, the probabilistic side of the problem and its random nature were considered [1]. In turn, in fuzzy sets, the observed states of cause–effect phenomena in the fuzzy conditions [15] (i.e., uncertainty as to the very fact of their occurrence) were analyzed.

The fact that the membership function of a fuzzy set assumes values from interval [0, 1] leads to the hasty conclusion that fuzziness is a hidden form of randomness, and therefore fuzzy set theory is basically nothing new in relation to probability. The differences between fuzziness and randomness, however, concern both their nature and the formal differences between probabilistic calculus and fuzzy sets. The nature of these phenomena lies in the problem of the uncertainty of the type of randomness and fuzziness. In the case of randomness, the event is strictly defined, while its occurrence is uncertain. Therefore, randomness can be equated with the uncertainty regarding an element's membership or non-membership. This is not the case with fuzziness, which concerns the very degree of membership of an element to a set, and therefore an event is no longer strictly defined. Such events are the ones analyzed in the paper—the occurring damage of building elements and the processes of their technical wear. Their nature, in the authors' opinion, is more fuzzy than random.

The differences between randomness and fuzziness can be presented with regard to the following three points of view:

- level of uncertainty;
- number of decision makers;
- number of steps in the decision process.

Regarding "the degree of uncertainty", the following decision-making situations, with an increasing degree of uncertainty, can be distinguished:

- Certainty: all the information that describes the issue of decision making is deterministic;
- Risk: information that describes the decision-making issue is probabilistic, i.e., the data have appropriate probability distributions;

- Uncertainty: even the probabilities are not known. Making decisions is usually reduced to using a minimax strategy;
- Fuzziness: uncertainty not only relates to the occurrence of an event, but also to its meaning in general, and this can no longer be considered using probabilistic methods. Of course, further extensions, such as adding risk to fuzziness, are also possible.

The sense of a fuzzy set can therefore be used to formally determine and quantitatively express ambiguous concepts that are always present in the programming and analysis of a construction process. Thus, fuzzy set theory is a theory of classes in which the transition from membership to non-membership does not have a jumping character, as is the case in a conventional set, but instead it is gradual. Striving for a quantification of criteria that are inherently qualitative (and therefore immeasurable), and trying to determine the relations between them, led to the use of the category of fuzzy sets with regard to this issue. Their properties enable elementary construction processes to be mathematically described as fuzzy events within an unambiguous quantitative (measurable) aspect.

To sum up, the approach of the creator of fuzzy set theory [8–10] (*Lofti Zadeh*, who, unlike *Yager* and *Kaufmann* [11,12], assumed the fuzzy set as a random event) was consciously used by the authors. This enabled the question of what is the probability that a building element is more or less (approximately) worn to be answered. Therefore, the differences between the concepts of fuzziness and randomness were not considered. It was only assumed that although these phenomena are different and described differently, they may nevertheless occur together as two types of uncertainty.

**Author Contributions:** Conceptualization, J.K., M.S. (Marek Sawicki) and M.S. (Mariusz Szóstak); methodology, J.K.; software, J.K. and M.S. (Mariusz Szóstak); validation, J.K., M.S. (Marek Sawicki) and M.S. (Mariusz Szóstak); formal analysis, J.K., M.S. (Marek Sawicki) and M.S. (Mariusz Szóstak); investigation, J.K. and M.S. (Marek Sawicki); resources, J.K. and M.S. (Marek Sawicki); writing—original draft preparation, J.K.; writing—review and editing, J.K. and M.S. (Mariusz Szóstak); supervision, M.S. (Marek Sawicki). All authors have read and agreed to the published version of the manuscript.

**Funding:** This research received no external funding.

**Institutional Review Board Statement:** Not applicable.

**Informed Consent Statement:** Not applicable.

**Data Availability Statement:** No new data were created or analyzed in this study. Data sharing is not applicable to this article.

**Conflicts of Interest:** The authors declare no conflict of interest.

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
