# Peer review of "Damage and Technical Wear of Tenement Houses in Fuzzy Set Categories"

_applsci, doi:10.3390/app11041484_

Round 1

Reviewer 1 Report

The main topic presented in the paper has a high potential of novelty related to the idea of quantify the relation between wear and maintenance. Despite this, the paper structure has to be improved.

In general, the paper seems to be focused on building aspects, speaking of damages and technical wear. The Fuzzy method is the tool through which this theme is analysed. However, the way in which the research is presented in the paper is very biased towards the method of analysis used, loosing the opportunity to emphasize the innovative achieved results. Just to give some examples, the main aim of the research and the methodology’s overview is presented only in the last paragraph. Up to that point it is not clear what the real purpose of the research and the operational expected results are.

A solid state of the art regarding the current knowledge about building’s wears is missed. Why could the use of the Fuzzy analysis improve the scientific background? Which is the upgrade given by the achieved results? How can I use the obtained information? Are results useful to address the maintenance strategies?

Some specific suggestions:

  • Lines 34-60: I suggest to summarise this entire text using the reference. In replacement of this, please add a framework related to the topic of building wares and damages.
  • Lines 61-73: the entire explanation is not necessary. The reference is enough to describe the concept. The paper is not focused on the Fuzzy analysis but it is about its use in the construction sector (add references).
  • Lines 79-93: the growing interest in Fuzzy analysis is known but, please, clarify how it was/is used in the construction sector. At lines 89-93 the reference is sufficient without the necessity to quote the title of the contribution and its description.
  • Lines 129-134: please, clarify the research problem, which is better explained in the last paragraph. Which are the problems given by the difficulties to quantify the wear’s criteria? It is supposed that the research could give new “tools” to substitute the visual analysis of damages, normally conducted by experts.
  • Line 160: specify “observed states”.
  • Line 150: I suggest to describe all the data, and their mutual relation, which enter the analysis system before describe in detail all the Fuzzy states. A greater correlation between analysed data and the case studies could be useful to understand the analysis. For example, the text block at lines 198-204 has to be described at the beginning as set up of the specific research to give the framework of the analysis. I suggest to invert paragraph 2.1 with 2.2 in order to introduce the research model before the Fuzzy set up.
  • Line 192: Indicate that this categorisation is made specifically for the case studies and the reason why, because it is not able to represent the entire group of wear’s types.
  • Lines 198-199: “The purpose of such a conceptual and technical systematization of damage is a comprehensive diagnosis of the extent to which a building element is worn”. Please clarify this sentence.
  • Lines 205-208: please explain how you obtained “Z”. Are you speaking regarding "average technical “condition of maintenance" or “technical wear of a building elements”? What does “semantic analysis” mean in this context?
  • Line 222: “middle” is not clear in relation to the 3 levels of maintenance.
  • Line 245: please specify the method of combination.
  • Lines 318-338: move this text to “Discussion” section.
  • Lines 339-385: move this part to “Result” paragraph. Line 359: very obvious concept. The one in brackets is more interesting; please give an example of it.
  • Lines 387-417: this text explain better then the introduction the aim and method of the research. The suggestion is to add this information at the beginning.

Author Response

Dear Reviewer of Applied Sciences,

Thank you for the review of our paper applsci-1097797 entitled “Damage and Technical Wear of Tenement Houses’ in Fuzzy Set Categories” to be published in the journal Applied Sciences, special issue “Probabilistic and Fuzzy Approaches for Estimating the Life Cycle Costs of Buildings”.

We appreciate your thoughtful and  accurate comments as well as appreciation of our research works. We have carefully considered all comments and have now completed the revisions incorporating  your suggestions in the revised uploaded manuscript.

We hope that the revised paper meets your expectations.

Kind regards,

JarosĹ‚aw Konior, Marek Sawicki and Mariusz Szóstak

Department of Building Engineering, Faculty of Civil Engineering, Wroclaw University of Science and Technology, 50-370 Wrocław, Poland

Here are answers to reviewer’s comments:

REVIEWER 1

General Comments. The main topic presented in the paper has a high potential of novelty related to the idea of quantify the relation between wear and maintenance. Despite this, the paper structure has to be improved. In general, the paper seems to be focused on building aspects, speaking of damages and technical wear. The Fuzzy method is the tool through which this theme is analysed. However, the way in which the research is presented in the paper is very biased towards the method of analysis used, losing the opportunity to emphasize the innovative achieved results. Just to give some examples, the main aim of the research and the methodology’s overview is presented only in the last paragraph. Up to that point it is not clear what the real purpose of the research and the operational expected results are. A solid state of the art regarding the current knowledge about building’s wears is missed. Why could the use of the Fuzzy analysis improve the scientific background? Which is the upgrade given by the achieved results? How can I use the obtained information? Are results useful to address the maintenance strategies?

Answers to General Comments. We appreciate the reviewer’s valuable view on the presented topic but cannot agree to a full extant. Probabilistic approach to the research has been previously developed by the authors’ in published papers (see added references [1-7]) which represent the current state of the art on buildings’ deterioration. However, randomness has fundamental limits while assessing technical state during technical inspections of tenement houses by experts with their natural subjective / descriptive way of measurement. Experts may not know if a hidden damage = event exists. Therefore fuzziness makes possible to convert qualitative appraisal of damages into quantitative / measurable ones. To clear up the purpose of research and background of research the further explanation has been added to item 1.3: “While appraising building elements’ technical wear - apart from applying the measurable (qualitative) criteria - the immeasurable (quantitative) criteria representing symptoms (pinpointed defects) of their deterioration have been taken into account. Only very few of these criteria can be classified at high level of probability. There are symptoms of extreme characters, described by extreme dichotomic divisions. It is, however, agreed that between e.g. a total pest attack to wooden elements and its lack the mid-states appear. Their value is often appreciate in verbal way, e.g. “substantially”, considerably”, “significantly”, “partially”, “hardly” and it is always met in a description of detected defects as a result of a building objects technical inspections. Therefore, the research led towards looking at the problem from the angle, which gave right to describe naturally qualitative variables (so immeasurable) and determine existing relations between them in fuzzy sets categories [15,25-39]. Advantages of fuzzy theory made possible to describe the defects, representing three middle states (II, III, IV) of conditions of the building elements’ maintenance, in a clear quantitative (measurable) aspect. Doubtless, fuzzy conditions are fully represented in these mid-states.” The upgrade of fuzzy approach has been laid out in item 3c: “Consideration of the problem with regards to fuzzy phenomena allows for the synthesis of elementary criteria. This gives the greatest approximations (at the stage of the technical investigation of a residential building) to the global assessment of the degree of wear of the building's elements. In addition, it significantly reduces the subjective factor of this assessment, which has the greatest impact on the result of research conducted for the middle maintenance states of buildings”. Usefulness of results and the maintenance strategy has been laid out in item 3b: “When determining the degree of damage of 10 selected building elements according to fuzzy criteria, it was indicated that there is a need for an individual approach to each of the elements (especially structural) during the process of their technical assessment. at the level of the greatest detail, the type of damage and the degrees of fuzzy damage to the elements of the downtown tenement houses were determined. In the most representative, i.e. average/satisfactory condition of maintenance - S (U) III - the degrees were determined as follows:

  • for foundations: brick decay 0.59
  • for basement walls: brick decrements 0.25
  • for solid floors above basements: brick decrements 0.22
  • for structural walls: mortar decrements 0.93
  • for wooden inter-storey floors: weeping 0.64
  • for internal stairs: mechanical damage 0.56
  • for roof constructions: weeping on wooden elements 0.43
  • for window joinery: mechanical damage 0.85
  • for inner plasters: plaster decay 0.85
  • for facades: cracks on plaster 0.94

The research – the methodological assumption, the mathematical procedure and the conclusions – should be treated as an exploratory work. Thus, it is an attempt of the recognition of the mechanism of the reasons and effects phenomena, which an engineer expert faces to while technically inspecting a building object. This assessment, however, is naturally gifted with an immeasurable aspect (partly subjective). Creating a new model of apartment houses’ technical inspection, based on procedures and conclusions drawn from the work, will make possible to transfer the weight of technical assessment from the qualitative form into quantitative one. Our intention is to direct the further work, connected with widely considered diagnoses of technical objects, towards the direction described in the paper.”

Some specific suggestions:

Comment 1. Lines 34-60: I suggest to summarise this entire text using the reference. In replacement of this, please add a framework related to the topic of building wares and damages.

Answer 1. References used. Framework related to damages and technical wear inserted the following way: “Essential research of tenement houses aims to undergo a qualitative analysis of detected defects and identify all particular defects of their elements. Therefore, the „reason - effect” model is applied as follows:”

Comment 2. Lines 61-73: the entire explanation is not necessary. The reference is enough to describe the concept. The paper is not focused on the Fuzzy analysis but it is about its use in the construction sector (add references).

Answer 2. This supplementary explanation was moved to discussion. References related to the construction sector added [27-39].

Comment 3. Lines 79-93: the growing interest in Fuzzy analysis is known but, please, clarify how it was/is used in the construction sector. At lines 89-93 the reference is sufficient without the necessity to quote the title of the contribution and its description.

Answer 3. Lines 89-93 deleted. The following clarification of fuzzy sets used in construction sectors has been added: “The stage preceding the main scope of the work was the conducting of a qualitative analysis of damage to the elements of the tested residential buildings [1,15]. The technical characteristics and typological ordering of this damage, understood as an expression of the quality of maintenance of residential buildings, enabled the exploitation conditions of the considered objects to be identified. A number of works by NowogoĹ„ska [16-20] were used in the methodical approach to the technical assessment of tenement houses, and the fuzzy calculus presented in the publications of Plebankiewicz, Wieczorek, and Zima [21-26] was used in the assessment of the whole service life of a building object. The works of Ibadov [27-30] and other authors [31-39], which concerned the construction investment process with the fuzzy phase, allowed for the practical application of uncertain and subjective events when deter-mining the degree of damage to the tested tenement houses.”

Comment 4. Lines 129-134: please, clarify the research problem, which is better explained in the last paragraph. Which are the problems given by the difficulties to quantify the wear’s criteria? It is supposed that the research could give new “tools” to substitute the visual analysis of damages, normally conducted by experts.

Answer 4. The research new tool stands for fuzziness which gives much greater possibilities of studying cause and effect relationships than the probabilistic analysis. The use of simple operations in the fuzzy set calculus enables for the simultaneous recognition of the impact of elementary damage that occurs with a specific frequency (probability), and also the measure of their interdependence (correlation) on the observed technical wear of building elements. As a result of the proposed model, which is based on fuzzy set theory, it is possible to identify the elementary damage that determines the degree of destruction of the building element. The result of the cumulative effects of frequently occurring mechanical damage to the structure and texture of elements indicates that this group of damage is no less important in the process of the technical wear of elements of downtown tenement houses. Consideration of the problem with regards to fuzzy phenomena allows for the synthesis of elementary criteria. This gives the greatest approximations (at the stage of the technical investigation of a residential building) to the global assessment of the degree of wear of the building's elements. In addition, it significantly reduces the subjective factor of this assessment, which has the greatest impact on the result of research conducted for the middle maintenance states of buildings

Comment 5. Line 160: specify “observed states”.

Answer 5. Observed states mean the real, investigated, inspected state of the buildings on the contrary to theoretical state assessed by theoretical formulas.

Comment 6. Line 150: I suggest to describe all the data, and their mutual relation, which enter the analysis system before describe in detail all the Fuzzy states. A greater correlation between analysed data and the case studies could be useful to understand the analysis. For example, the text block at lines 198-204 has to be described at the beginning as set up of the specific research to give the framework of the analysis. I suggest to invert paragraph 2.1 with 2.2 in order to introduce the research model before the Fuzzy set up.

Answer 6. We are sorry but cannot agree with the suggestion as being not very much sensible. Background of fuzzy sets should come prior to the research model which relays to fuzziness and is literary related to fuzzy algorithms. What is more, all formulas are explained in logical steps in item 2.1 as to be applied in item 2.2.

Comment 7. Line 192: Indicate that this categorisation is made specifically for the case studies and the reason why, because it is not able to represent the entire group of wear’s types.

Answer 7. Good point. The sentence in the line 192 has been particularised the way that indicates categorisation reference to the researched group of buildings: “When visually assessing the technical wear of building elements that inspected tenement houses are consist of, the symptoms of their damage are taken into account.”

Comment 8. Lines 198-199: “The purpose of such a conceptual and technical systematization of damage is a comprehensive diagnosis of the extent to which a building element is worn”. Please clarify this sentence.

Answer 8. Let’s say the other words: The proposed systematization of damages and their split into 4 typological groups is made for easier diagnosis of elements’ defects until they are completely worn.

Comment 9. Lines 205-208: please explain how you obtained “Z”. Are you speaking regarding "average technical “condition of maintenance" or “technical wear of a building elements”? What does “semantic analysis” mean in this context?

Answer 9. This is clearly expressed: “It is difficult to define a fuzzy set with such a broad meaning as "average technical condition of maintenance" using one membership function. In this case, a semantic analysis of the term "technical wear of a building element" was used, which was denoted with the symbol of a fuzzy set "Z". So “Z” is not an "average technical condition of maintenance"; “Z” stands for "technical wear of a building element".

Comment 10. Line 222: “middle” is not clear in relation to the 3 levels of maintenance.

Answer 10. There are 5 technical condition of maintenance (I – V). Middle maintenance states are equivalent to a satisfactory (II), average (III) and poor (IV) technical condition of maintenance where fuzzy damages are the best representative (not as extreme as in states I and II).

Comment 11. Line 245: please specify the method of combination.

Answer 11. 30 detected elementary defects u1 - u30 have been combined with pairs or triples of defects related to different elements but the same type, e.g. {u15, u16, u23} Û dampness, weeping, and flooding with water.

Comment 12. Lines 318-338: move this text to “Discussion” section.

Answer 12. Pre-conclusions in lines 318-338 moved to “Discussion” as suggested.

Comment 13. Lines 339-385: move this part to “Result” paragraph. Line 359: very obvious concept. The one in brackets is more interesting; please give an example of it.

Answer 13. Lines 339-385 are in “Result” paragraph. The second title “Conclusions” of that section was deleted as not mislead the reader. It is generally known that the degree of damage to the element increases with the deterioration of its maintenance conditions but it was calculated by application of fuzzy formulas. For instance degrees of fuzzy damage set S(U) corresponding to the maintenance states II, III and IV are growing the following way: Z3 – basement walls - u3 – brick losses: 0.05; 0.25; 0.67

Comment 14. Lines 387-417: this text explain better than the introduction the aim and method of the research. The suggestion is to add this information at the beginning.

Answer 14. Good point. The explanation regarding the aim of the research was added at the beginning of the article.

Reviewer 2 Report

There are no comments but it is necessary to check and improve the English language which is not always understandable. The references should be enriched with some indication to previous research.

Author Response

Dear Reviewer of Applied Sciences,

Thank you for the review of our paper applsci-1097797 entitled “Damage and Technical Wear of Tenement Houses’ in Fuzzy Set Categories” to be published in the journal Applied Sciences, special issue “Probabilistic and Fuzzy Approaches for Estimating the Life Cycle Costs of Buildings”.

We appreciate your thoughtful and  accurate comments as well as appreciation of our research works. We have carefully considered all comments and have now completed the revisions incorporating  your suggestions in the revised uploaded manuscript.

We hope that the revised paper meets your expectations.

Kind regards,

JarosĹ‚aw Konior, Marek Sawicki and Mariusz Szóstak

Department of Building Engineering, Faculty of Civil Engineering, Wroclaw University of Science and Technology, 50-370 Wrocław, Poland

Here are answers to reviewer’s comments:

REVIEWER 2

General Comments. There are no comments but it is necessary to check and improve the English language which is not always understandable. The references should be enriched with some indication to previous research.

Answers to General Comments. English language has been rechecked and improved to the higher level of technical language and its comprehensiveness for the reader. Also the references have been enriched with plenty of crossrefs to previous research works on the topic of the main author and recent inputs of co-authors – as follows:

  • Konior, J. Decision assumptions on building maintenance management. Probabilistic methods, Civ. Eng. 2007, 53, 403–423.
  • Konior, J. Technical assessment of old buildings by fuzzy approach, Archives of Civil Engineering, 2019, 65(1), pp.129–142, https://doi.org/10.2478/ace-2019-0009.
  • Konior, J. Technical Assessment of old buildings by probabilistic approach., Archives of Civil Engineering, 2020, 66(3), pp. 443–466, https://doi.org/https://doi.org/10.24425/ace.2020.134407.
  • Konior, J. Maintenance of apartment buildings and their measurable deterioration, Trans. Czas. Tech. 2017, 6, 101–107, https://doi.org/10.4467/2353737xct.17.090.6566.
  • Konior, K. Bi-serial correlation of civil engineering building elements under constant technical deterioration, Sci. Gen. Tadeusz Kosiuszko Mil. Acad. L. Forces. 2016, 179, 142–150.
  • Konior, J. Intensity of defects in residential buildings and their technical wear, Tech. Trans. Civ. Eng. 2014, 111(2-B), 137–146.
  • Konior, J.; Sawicki, M.; Szóstak, M. Intensity of the Formation of Defects in Residential Buildings with Regards to Changes in Their Reliability.  Sci.202010, 6651. doi: 10.3390/app10196651
  • Konior, J.; Sawicki, M.; Szóstak, M. Influence of Age on the Technical Wear of Tenement Houses. Sci. 2020, 10, 6651. doi: 10.3390/app11010297

Reviewer 3 Report

Lines 409-454 do a good job of distinguishing fuzziness from randomness and probability theory; although, Zadeh's possibility theory and his computing with words should be added. Why base the studies of aging buildings on fuzzy logic, when CW provides a more natural descriptive mechanics? I'd like to see this question addressed and the response made part of the paper proper. For example, in the decay of plaster you mention, we are more concerned with finding the underlying model than mapping the rate of decay into a fuzzy set model. A qualitative model is suggested. This is supported by CW. Of course, the numerical portions of such models could be represented using Type 2 fuzzy functions. But, then how do you find for the underlying qualitative aspects? We should not rewrite the paper. Rather, I suggest expanding it with a section on computing with words nd how one can use CW to find for the qualitative model. Then, compare and contrast the use of fuzzy logic with CW so readers will acquire a better understanding of which formalism might be better suited for their application domain. You can also add a section on Type 2 fuzzy logic to support your use of fuzzy logic for the numerical modeling. Update your intro and conclusion with this information and your paper should then be publishable. I expect this approach to generate ideas for a follow-on paper.

Author Response

Dear Reviewer of Applied Sciences,

Thank you for the review of our paper applsci-1097797 entitled “Damage and Technical Wear of Tenement Houses’ in Fuzzy Set Categories” to be published in the journal Applied Sciences, special issue “Probabilistic and Fuzzy Approaches for Estimating the Life Cycle Costs of Buildings”.

We appreciate your thoughtful and  accurate comments as well as appreciation of our research works. We have carefully considered all comments and have now completed the revisions incorporating  your suggestions in the revised uploaded manuscript.

We hope that the revised paper meets your expectations.

Kind regards,

JarosĹ‚aw Konior, Marek Sawicki and Mariusz Szóstak

Department of Building Engineering, Faculty of Civil Engineering, Wroclaw University of Science and Technology, 50-370 Wrocław, Poland

Here are answers to reviewer’s comments:

REVIEWER 3

General Comments. Lines 409-454 do a good job of distinguishing fuzziness from randomness and probability theory; although, Zadeh's possibility theory and his computing with words should be added. Why base the studies of aging buildings on fuzzy logic, when CW provides a more natural descriptive mechanics? I'd like to see this question addressed and the response made part of the paper proper. For example, in the decay of plaster you mention, we are more concerned with finding the underlying model than mapping the rate of decay into a fuzzy set model. A qualitative model is suggested. This is supported by CW. Of course, the numerical portions of such models could be represented using Type 2 fuzzy functions. But, then how do you find for the underlying qualitative aspects? We should not rewrite the paper. Rather, I suggest expanding it with a section on computing with words and how one can use CW to find for the qualitative model. Then, compare and contrast the use of fuzzy logic with CW so readers will acquire a better understanding of which formalism might be better suited for their application domain. You can also add a section on Type 2 fuzzy logic to support your use of fuzzy logic for the numerical modeling. Update your intro and conclusion with this information and your paper should then be publishable. I expect this approach to generate ideas for a follow-on paper.

Answers to General Comments. We appreciate the reviewer’s valuable view on the presented topic but cannot agree to a full extant in terms of neglecting computing aspects of the research works. Lofti Zadeh [8-10] presented the main and basic ideas of soft computing in his paper “Soft Computing and Fuzzy Logic” published in 1994 which is known to the authors. Due to this theory many other commercial, scientific and consumer appliances were introduced and came in global markets. The hybridized fuzzy techniques proved much better and result oriented in real life as those have more fault tolerance, so there is a lot of done in these fields for further applications to make human life easier and to benefit public and societies. However, we are rather focused on seeking alliances among damages, technical wear and maintenance conditions of the local tenement houses which are facing decisions of demolishing vs. repair.  Therefore, we are closer to approach laid out in the monograph of Kasprzyk [14] "Fuzzy sets in system analysis" where the author presented the application of fuzzy sets in the broadly understood system analysis - models of decision making and control. Why base the studies of aging buildings on fuzzy logic, when CW provides a more natural descriptive mechanics? Because CW approach has been already recognised within the same research sample and presented in serious of references [1-7]. A qualitative model is not suggested. What is proposed it is natural transformation of qualitative model into a quantitative one. So, this is what we need fuzzy set theory apparatus for. In the presented research results the degrees of fuzzy damage set S(U) corresponding to the maintenance states II, III and IV were determined in quantitative way which is priceless. As to achieve the goal of our pretty complex research (impossible get a building laboratory tested!)  we selected the most straightforward fuzzy tools: membership function, conditional probabilities of Bayes, fuzzy relations and matrixes. The output of these findings was laid out in the paper [15] and based on the latest reissue of fuzzy fundamentals: 33.40.           Zadeh L.; Aliev, R. Fuzzy Logic Theory and Applications, World Scientific Publishing Co Pte Ltd., 2018, Singapore [10]. To sum up, the following clarification has been inserted in the paper, item 1.3. “While assessing building elements’ technical wear - apart from applying the measurable (qualitative) criteria - the immeasurable (quantitative) criteria representing symptoms (pinpointed defects) of their deterioration have been taken into account. Only very few of these criteria can be classified at high level of probability. There are symptoms of extreme characters, described by extreme dichotomic divisions. It is, however, agreed that between e.g. a total pest attack to wooden elements and its lack the mid-states appear. Their value is often appreciate in verbal way, e.g. “substantially”, considerably”, “significantly”, “partially”, “hardly” and it is always met in a description of detected defects as a result of a building objects technical inspections. Therefore, the research led towards looking at the problem from the angle, which gave right to describe naturally qualitative variables (so immeasurable) and determine existing relations between them in fuzzy sets categories [8-15]. Advantages of fuzzy theory made possible to describe the defects, representing three middle states (II, III, IV) of conditions of the building elements’ maintenance, in a clear quantitative (measurable) aspect. Doubtless, fuzzy conditions are fully represented in these mid-states.” Having said that, we are asking the reviewer to accept our point of view with thorough explanation and let us generate suggested valuable ideas for follow-on papers.

Reviewer 4 Report

The paper shows generalization of the assessment of the building element’s technical wear by applying developed method based on the fuzzy set theory. The paper is well structured and presents all the needed information for reader to understand results and conclusions, no matter if reader is an expert in this area or an amateur. From my point of view, manuscript requires no additional changes and I advise that it can be accepted in present form.

Author Response

Dear Reviewer of Applied Sciences,

Thank you for the review of our paper applsci-1097797 entitled “Damage and Technical Wear of Tenement Houses’ in Fuzzy Set Categories” to be published in the journal Applied Sciences, special issue “Probabilistic and Fuzzy Approaches for Estimating the Life Cycle Costs of Buildings”.

We appreciate your thoughtful and  accurate comments as well as appreciation of our research works. We have carefully considered all comments and have now completed the revisions incorporating  your suggestions in the revised uploaded manuscript.

We hope that the revised paper meets your expectations.

Kind regards,

JarosĹ‚aw Konior, Marek Sawicki and Mariusz Szóstak

Department of Building Engineering, Faculty of Civil Engineering, Wroclaw University of Science and Technology, 50-370 Wrocław, Poland

Here are answers to reviewer’s comments:

REVIEWER 4

General Comments. The paper shows generalization of the assessment of the building element’s technical wear by applying developed method based on the fuzzy set theory. The paper is well structured and presents all the needed information for reader to understand results and conclusions, no matter if reader is an expert in this area or an amateur. From my point of view, manuscript requires no additional changes and I advise that it can be accepted in present form.

Answers to General Comments. Thank you for your openness and appreciation of our research works as well as the fuzzy approach. While assessing building elements’ technical wear - apart from applying the measurable (qualitative) criteria - the immeasurable (quantitative) criteria representing symptoms (pinpointed defects) of their deterioration have been taken into account. Only very few of these criteria can be classified at high level of probability. There are symptoms of extreme characters, described by extreme dichotomic divisions. It is, however, agreed that between e.g. a total pest attack to wooden elements and its lack the mid-states appear. Their value is often appreciate in verbal way, e.g. “substantially”, considerably”, “significantly”, “partially”, “hardly” and it is always met in a description of detected defects as a result of a building objects technical inspections. Therefore, the research led towards looking at the problem from the angle, which gave right to describe naturally qualitative variables (so immeasurable) and determine existing relations between them in fuzzy sets categories [8-15].

Round 2

Reviewer 1 Report

I think the paper is now suitable for the publication, after the provided reviews.